# A supply and demand intervention increased fish consumption among rural women: A randomized, controlled trial

Alexander Tilley[1]*, Kendra A. Byrd[2], Hamza Altarturi[1], Gianna Bonis-Profumo[1], Joctan Dos Reis Lopes[1], Mario Gomes[1], Katherine Klumpyan[3], Lorenzo Longobardi[1], Kelvin Mashisia Shikuku[4]

**1** WorldFish, Penang, Malaysia, **2** College of Health and Human Sciences, San Jose State University, San Jose, California, United States of America, **3** Mercy Corps, Timor-Leste, **4** International Livestock Research Institute, Nairobi, Kenya

☯ These authors contributed equally to this work

* alex.tilley@gmail.com

## Abstract

Malnutrition is a critical public health issue in Timor-Leste, where nearly half of children under five suffer from stunting and diets are chronically low in nutrient-rich foods, including fish. We conducted a cluster-randomized, 2 × 2 factorial, parallel-arm controlled trial to evaluate the effects of nearshore fish-aggregating devices (FADs, a technology designed to increase pelagic fish catch rates), social behaviour change (SBC) interventions, and their combination, on household fish purchasing and consumption in inland villages of Timor-Leste. However, inland households exposed to both FADs and SBC were nearly twice as likely to purchase fish (PR: 1.90, 95% CI: 1.14–3.20, p < 0.05) and women were over four times as likely to report fish consumption the previous day (PR: 4.17, 95% CI: 1.88–9.29, p < 0.001), compared to controls. No significant effects were observed from FADs or SBC alone. These findings suggest that the combination of supply-side (FAD) and demand-side (SBC) interventions is necessary to improve dietary intake in nutritionally vulnerable, inland populations. These results underscore the importance of integrated food system approaches to address poor diet quality and reduce malnutrition risks in small island developing states. Trial registration: Trial registered at clinicaltrials.gov Identifier: NCT04729829.

## 1. Introduction

Current food systems are unsustainable and inequitable, perpetuating high rates of malnutrition and environmental destruction [1]. Fish and other aquatic foods are gaining attention for their potential to efficiently provide two fundamental components of sustainable, nutritious food systems – higher nutrient yield with fewer environmental inputs [2,3]. Small pelagic fish, especially, are a sustainable, affordable, and nutritious

**Data availability statement:** The new data availability statement for inclusion is as follows: Code and data supporting the analyses are available via Zenodo (DOI: https://doi.org/10.5281/zenodo.18166806) and the associated GitHub repository (https://github.com/WorldFishCenter/timor.fads).

**Funding:** This work was carried out under the Fisheries Sector Support Program – Phase 2, funded by the Royal Norwegian Embassy in Jakarta. This publication was supported by the CGIAR Aquatic Foods Initiative led by WorldFish and UK International Development as part of the Asia–Africa BlueTech Superhighway Project led by WorldFish (FCDO Project Grant Number: 301203). The funders provided support in the form of salary contributions for authors and field costs but did not have any additional role in the study design, data collection and analysis, decision to publish, or preparation of the manuscript.

**Competing interests:** The authors have declared that no competing interests exist.

option for addressing nutritional deficiencies in women, children, and vulnerable populations, particularly when consumed whole [4–6]. However, the benefits of fish are often concentrated near fisheries as people who live near fisheries, including young children [7], benefit most from them [8]. Distributing fish to villages, towns, or cities that are further from water bodies remains a challenge, even, in some cases, when the fish are dried [9].

Malnutrition remains a persistent and severe public health challenge in Timor-Leste, with one of the highest rates of child stunting in the world [10]. Nationally, nearly half of children under five are stunted, and women of reproductive age experience widespread micronutrient deficiencies, driven largely by monotonous diets low in animal-source foods [11]. Fish, a nutrient-dense and culturally acceptable food in Timor-Leste, offers a promising solution to address these gaps, particularly as it provides bioavailable nutrients essential for growth, immunity, and cognitive development [4,5].

Despite being a coastal nation, access to fish in Timor-Leste is highly inequitable. While per capita fish consumption in coastal areas reaches approximately 17 kg/year, inland populations—where most of the population lives—consume as little as 4 kg/year [12,13]. Physical isolation, poor road infrastructure, and weak supply chains contribute to this disparity, limiting the availability of fish in upland markets and exacerbating nutritional inequities between rural and coastal communities [14].

Expanding the reach of small-scale fisheries (SSF) to inland populations has been proposed as a nutrition-sensitive strategy to improve diet quality [6]. One such intervention is the use of nearshore fish aggregating devices (FADs), which are low-cost, climate-adaptive technologies designed to increase the catchability of pelagic species by concentrating fish around floating structures [15]. However, while FADs may boost supply [16,17], their effectiveness in improving nutrition outcomes depends on whether increased availability translates into greater household access and consumption.

At the same time, behavioural and social factors influence whether fish, once available, are purchased and consumed [18]. In Timor-Leste, cultural beliefs, taboos, and limited awareness about the nutritional benefits of fish, particularly for young children and pregnant women, can limit demand [19]. Increasing women's dietary intake and nutrition knowledge has been shown to improve overall household food choices and is strongly associated with better child health and nutrition outcomes, particularly in the critical first 1,000 days [20]. Social and behaviour change (SBC) communication, tailored to local contexts, can address these barriers by promoting positive dietary practices with women [21,22]. However, actively involving men in discussions and awareness-raising of better diets in rural communities, can also have direct improvements in household nutrition [23–25]. This is particularly relevant in Timor-Leste, where men often make final decisions on household food purchases, despite women being primarily responsible for meal preparation [26].

To test the effectiveness of an integrated food systems approach, we conducted a cluster-randomized, 2 × 2 factorial controlled trial in inland Timor-Leste [27]. The study evaluated whether pairing FADs (a supply-side intervention) with SBC activities (a demand-side intervention) would increase fish purchase by households and

consumption among women in rural households. To the best of our knowledge, this is the first randomized controlled trial investigating a nutrition-sensitive fisheries intervention.

We hypothesized that households receiving both FADs and SBC interventions would exhibit significantly higher rates of household fish purchase and individual fish consumption among women, compared to households receiving either intervention alone or no intervention.

## Methods

### 2.1 Study site and selection of the municipalities

Timor-Leste is considered a small island developing state by the UN [28], and is predominantly Catholic (>90%), but with >60% of those declaring as Catholic following Indonesian occupation and the occupying government's insistence of approved theistic religions [29].

Six municipalities of Timor-Leste (Bobonaro, Covalima, Dili, Liquiça, Manatuto and Manufahi) (Fig 1) were purposively selected based on the coverage of Mercy Corps programming of Village Savings and Loan Association (VSLA) groups, and the fact that each municipality had both coastal and inland villages. Dili is the smallest municipality in area but has the highest population density due to the capital city (Table 1). All municipalities except for Dili are predominantly pastoral, with low population density and high dependence on agricultural livelihoods for income and subsistence (Table 1). Just over 3% percent of households identify as fishers [31]. Using WorldFish and government data on fisheries landing sites in the six municipalities, rural coastal fishing villages with over 10 active fishing vessels were listed. Villages that already utilised FADs, or where no established trade routes taking fish inland existed, were excluded because no incentives or interventions were planned with distributors. Baseline focus groups with 3–6 traders encountered in each coastal village were used to establish existing trade routes, which predominantly aligned with the road network.

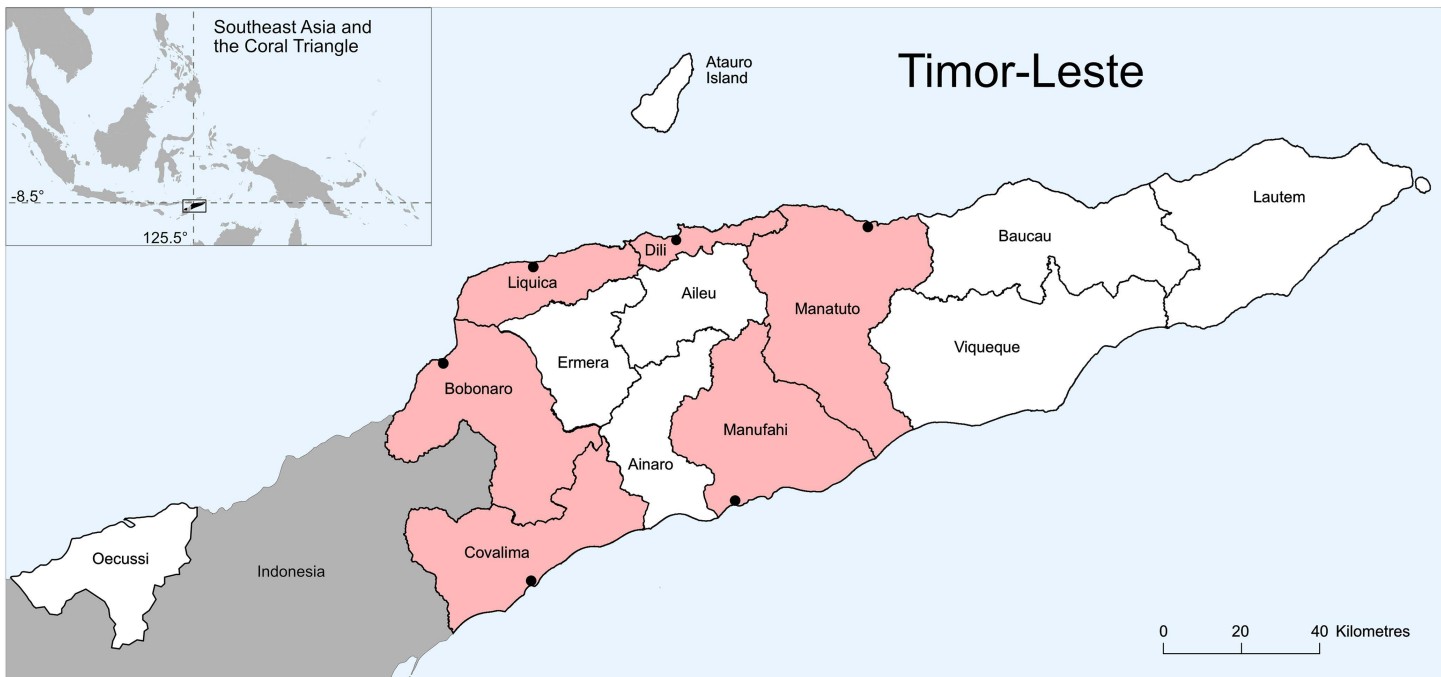

**Fig 1. Map of Timor-Leste.** Pink shading represents municipalities used in this study. Reprinted from Tilley et al. (2022) under a CC BY license, with permission from the author, original copyright 2022.

**Table 1. Summary area and demographic information on the six municipalities in this study from Timor-Leste 2015 national census [30].**

| Municipality | Area (km) | Households (n) | Households involved in agriculture | Population | Population density (ca/km²) |
|---|---|---|---|---|---|
| Bobonaro | 1,376 | 17,635 | 98% | 98,932 | 72 |
| Covalima | 1203 | 12,564 | 98% | 64,550 | 54 |
| Dili | 367 | 42,485 | 61% | 252,884 | 137* |
| Liquiça | 549 | 11,885 | 98% | 73,027 | 133 |
| Manatuto | 1782 | 7,467 | 95% | 45,541 | 26 |
| Manufahi | 1323 | 9,023 | 99% | 52,246 | 39 |

**\*Note:** Rural population density. 80% of the municipal population reside in the capital city of Dili. Including the urban population returns a municipal population density of 689ca/km².

## 2.2 Treatment assignment

The study is a cluster-randomized, 2×2 factorial, parallel-arm controlled trial. Randomization was by Excel-generated random numbers; allocation was concealed at village level. Masking was maintained for statistical analyses as far as feasible.

Six coastal and geographically dispersed villages (one per municipality) were chosen to minimize the risk of contamination between treatment and control sites. Contamination was minimised for non-FAD sites with fish caught at a FAD. The six coastal villages were randomly assigned to one of two experimental arms, namely FAD (n = 4) and non-FAD (n = 2). Two fish aggregating devices (FADs) were deployed in the nearshore fishing grounds of each coastal fishing village with already established trading links to inland study sites (Fig 2). The FAD treatment arm is imbalanced 2:1 because although the efficacy of nearshore FADs at increasing catch rates of fish has been established, it is also shown to be dependent upon local social, ecological and bathymetric conditions, and thus under some conditions, FADs do not increase catch [17]. B installing more FADs, we increased the chances of some of the sites showing higher catch rates.

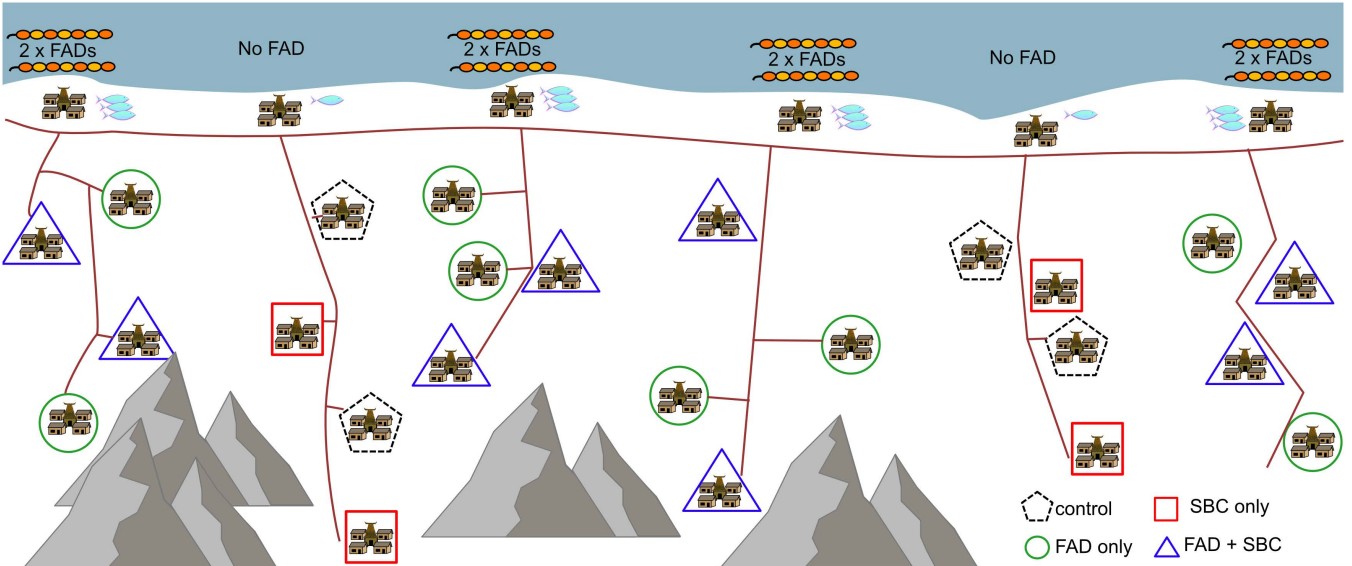

**Fig 2. A sketch representation of the two treatment levels of the randomised controlled trial.** 1. Coastal nearshore fish aggregating devices and 2. Social and behaviour change activities in rural inland communities in Timor-Leste. This diagram is a visualization only and does not represent the location of villages in the study. Adapted from ©2022 Tilley et al. Reproduced with permission.

## 2.3 Description of the FAD intervention to enhance fish supply

The design, construction and deployment of nearshore FADs used in this study were based on work carried out by World-Fish in partnership with the General Directorate of Fisheries in Timor-Leste since 2013 (Tilley et al. 2019). The FAD is made up of a series of buoys and attractant netting attached to nylon (sinking) and polypropylene (floating) ropes that are moored to the seabed using concrete blocks and a grapple anchor (Fig 3), known as an Indo-Pacific FAD [32]. FADs were constructed and deployed with local communities at the four treatment sites in September and October 2021, with their deployment location guided by fishers' knowledge of the area combined with bathymetric mapping for the seabed using a Furuno GP1670F depth sounder (Fig 3).

Upon first visiting the communities randomly selected to receive the FADs treatment, it was explained to the community that FADs were available and open to all fishers and were not the property of any individual or fishing group. FAD placement location and depth was decided through consultation with local fishers and leaders. FADs were checked and maintained by WorldFish staff every 6 months, which involved checking the FAD was still present and floating, and cleaning biofouling algae and encrusting organisms from the buoys, ropes and swivels to a depth of 5m to maintain maximum buoyancy of the FAD.

## 2.4 Inclusion/exclusion criteria of SBC villages

To be included as one of the inland villages in the study, the village had to have a VSLA already established and be located within 30 km inland of the coastal sites chosen as FAD and non-FAD sites. A further inclusion criterion for the villages was that fish traders from the coastal landing sites confirmed through interview that they sold fish products to those respective villages. 24 inland villages were selected and randomly assigned to one of two experimental arms namely SBC (N = 12) and non-SBC (N = 12). Villages were randomly allocated to treatment or control arms using an Excel random number table to a 1:1 allocation ratio. This established four treatment groups of Control (no FAD or SBC), FAD only, SBC only and FAD + SBC (Fig 1). This study employed a parallel design. Village participants and Mercy Corps staff as implementers of SBC interventions, were masked to the assignment of villages to FAD/non-FAD treatments. Study villages were all inland from the coast, and no supply chain interventions were carried out, so no evidence for downstream treatment assignation was available to village participants.

## 2.5 Description of SBC intervention to enhance demand for fish

The social and behaviour change (SBC) intervention was guided by the theory of planned behaviour and the socio-ecological model [33]. Information, education and communication (IEC) materials were designed using best practices in SBC [25]. All IEC materials were developed for low literacy populations and field tested before implementation. Between February 2021 and January 2022, within each SBC treatment village, a sequenced implementation of four key lessons (1. Nutrition, 2. Importance of protein and the benefits of consuming fish, 3. Gender awareness and resource allocation, and 4. Fish handling, hygiene, and cooking demonstration) was run with the VSLA groups. A reality style, interactive fish promotion video with a choose-your-own ending, reinforced content on the key themes. The VSLA members participated in skill-building activities such as fish deboning, cooking demonstrations and budgeting for nutritious family meals. Activities allowed them to practice the key promoted behaviours about fish nutrition and household decision-making on fish consumption. Key promoted behaviours included: 1) Including fish in family meals at least twice a week; 2) Picking bones out of fish for small children to start offering fish to infants at 6 months of age; 3) Having household conversations on the importance of allocating resources for protein purchases, including fish; and 4) Targeting money toward increasing protein consumption for families, focusing on fish.

In treatment villages (N = 12), larger scale SBC activities were also conducted to include the wider community and leaders to create a whole of community enabling environment for change and promote increased fish consumption.

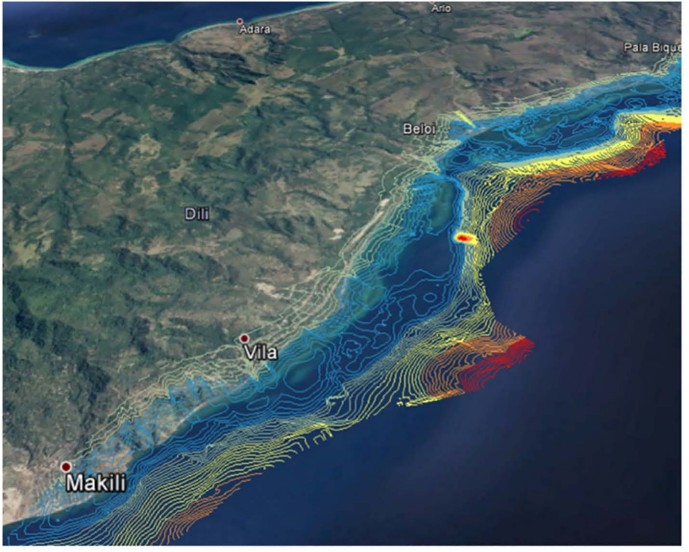

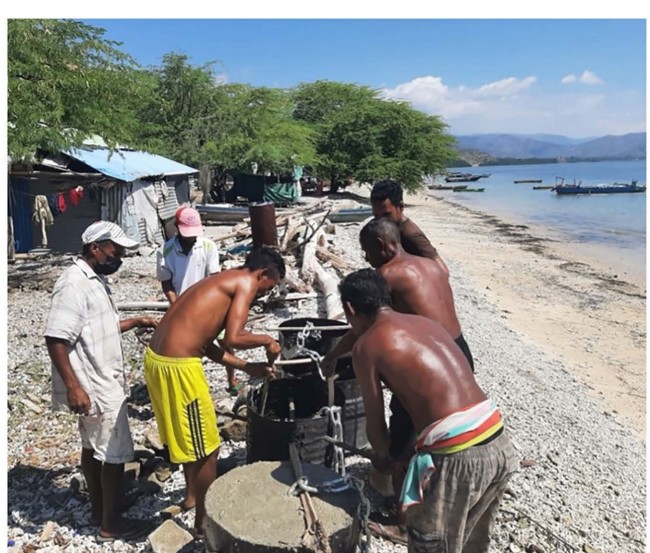

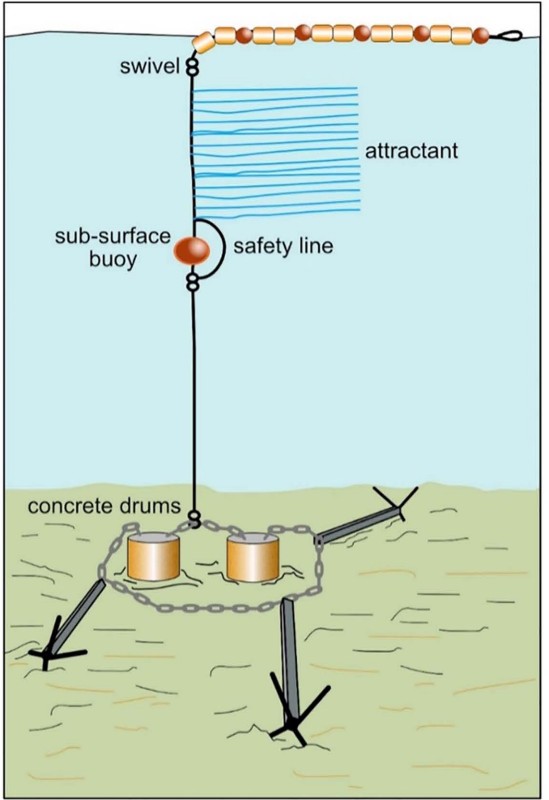

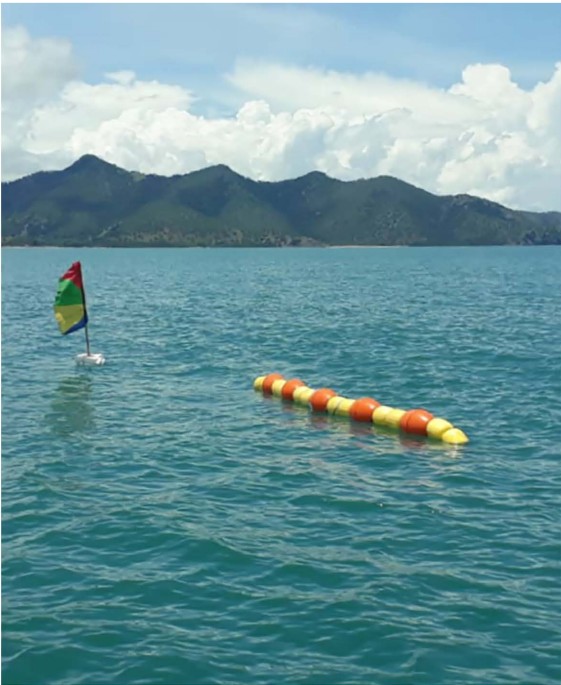

**Fig 3. Top Left: A contour map of bathymetry created using depth measurements of the seabed in Timor-Leste Created using ArcGIS® software by Esri.** Copyright © Esri. All rights reserved. Top Right: Fishers helping to construct the FAD anchors in Hera, Timor-Leste. ©WorldFish/Mario Pereira. Bottom Left: A diagram of a nearshore fish-aggregating device (FAD) (not to scale) ©2022 Tilley et al. Reproduced with permission. Bottom Right: The surface marker buoys and marker flag of a fish aggregating device deployed off the south coast of Timor-Leste in 2021. ©WorldFish/Mario Pereira.

Activities targeting inclusion of men (especially those with decision-making power) has been shown to be a strong design element for enabling change, especially for purchase/consumption of animal source foods [23,24]. Activities included a broadcasted radio drama reinforcing content from the fish promotion video, fish promotion activities on market day with collaboration from fish vendors who promoted their product, *Edu-tainment* competitions at schools to include nutrition and hygiene games for students, and a cooking competition amongst adults. The contestants prepared fish recipes and winners were determined based on hygienic food preparation, taste, and knowledge of the benefits of fish consumption as part of a balanced diet. SBC activities emphasized fish consumption for pregnant and lactating mothers, and children under 5 years of age. The control villages (N = 12) did not receive the nutrition campaign package.

### 2.6 Data and construction of variables

**2.6.1 Quantifying the effect of FADs on fish supply.** Fishers' catches were recorded daily at each of the landing sites by an enumerator using the 'Peskas' catch monitoring system on a tablet (www.timor.peskas.org). Peskas is a data and analytical workflow that connects open-source programs to collect, communicate, analyse and visualise small-scale fisheries movement and catch data on a dashboard [34,35].

The number and length of fish and other aquatic foods (crabs, octopus, lobster) were recorded in species, family and functional groups used for management purposes (S1 Table). The catch per unit effort (CPUE) was calculated in kg/fisher*hr (i.e., the weight of fish landed in kg, divided by number of fishers on the boat and the number of hours and the stated duration of the fishing trip). The average CPUE per landing site was calculated to ensure a fair comparison (irrespective of the habitat, gear type or boat type used for the fishing trip).

**2.6.2 Calculating fish consumption per household member.** Recruitment for the study was conducted concurrently with deployment of the baseline survey and ran between 3rd February 2021 and 25th September 2021, given several interruptions due to COVID. The sampling unit was the household in inland villages. Households were selected using simple random sampling from VSLA rosters or village household listings, and the respondent within the household was the primary food purchaser, typically an adult woman. Participation was voluntary and participants provided their written consent in the form of a signature documented digitally, having read or listened to the full details of the study and interventions. At endline, the surveys took place in August and September of 2022. To calculate the grams of fish consumed per household member per week, respondents were asked if a household member had purchased any fish or aquatic food in the previous seven days. If so, the name of the fish was recorded in Tetum by the enumerator (or selected from a multiple-choice menu) and the amount of fish purchased was reported in kilograms (if known) or by length in centimetres. Where length was reported, the weight in kg was calculated using species specific length-weight conversion parameters from FishBase [36]. Smoked and dried fish were calculated in the same way, along with a multiplier to account for the average edible portion of finfish at 87% [37,38]. If the length and weight variables were not found in FishBase, or if households reported purchasing fish pieces or other aquatic foods, length and weight measurements were collected from the local market. We also did this for fish that were not purchased fresh (i.e., they were purchased smoked or dried). The market fish were weighed in grams using a kitchen scale (Soehnle 61507), or in kg using the market scale. Length was taken using a measuring tape, and centimetres were measured to the 0.1 place. The weight of the canned fish was recorded based on the type of canned fish reported given that the two types (sardines and tuna) are sold in specific sizes. Once the total grams of fish and aquatic foods purchased by the household was calculated, the grams were divided by the members of the household to provide the apparent weekly fish consumption per capita.

**2.6.3 Dietary and feeding practice variables.** We conducted a qualitative 24-hour recall of the woman head of the household to calculate dietary diversity, which includes fish and aquatic food consumption. Dietary diversity was assessed using a qualitative 24-hour recall following FAO's Minimum Dietary Diversity for Women guidelines [39]. For a child in the household between 6 and 24 months old, we asked the caregiver responsible for feeding the child if they had been given any fish or other aquatic foods the day prior.

## 2.7 Statistical methods

Sample size calculations assumed a minimum detectable effect of 15%, 0.05 intra-cluster correlation coefficient, and 80% power. To accommodate 25% attrition, we aimed for 720 households (about 30 per village across 24 sites). The intra-cluster correlation coefficient (ICC) of 0.05 used in our sample size calculations is consistent with empirical ICC values reported in nutrition-sensitive agricultural and health behaviour cluster trials conducted in low- and middle-income countries. Prior studies in similar community-based interventions—particularly those measuring dietary behaviour outcomes—have used ICC estimates ranging between 0.01 and 0.10, with 0.05 being a commonly adopted mid-range conservative value to account for moderate within-cluster correlation [40–42]. Given that our primary outcomes involved household-level dietary behaviours influenced by shared environmental and cultural contexts within villages, we considered an ICC of 0.05 to be a reasonable and cautious estimate for power calculation.

To assess the impact of Fish Aggregating Devices (FADs) on fish catch per unit effort (CPUE), we used a difference-in-difference (DiD) approach to estimate changes in CPUE before and after FAD installation at treatment vs. control sites. We defined equal pre/post timeframes based on FAD duration per site and used bootstrapping (1,000 replicates) to construct confidence intervals. To delineate the before and after periods for each site, we segmented the data into two equal timeframes corresponding to the maximum duration FADs were present at each location, as outlined in Table 2. For example, at Atabae (Bobonaro), FADs endured for 581 days; hence, we examined the CPUE from the 581 days preceding installation against the 581 days following it. An identical temporal comparison was applied at Hera (Dili) and Suai (Covalima), which had FADs for 657 and 119 days, respectively. This stratification allowed us to isolate the specific effect of FADs on CPUE, while accounting for potential confounders such as seasonal effects or fish population trends.

Baseline differences between treatment arms were tested using adjusted linear or logistic models. Variables differing at $p < 0.05$ were included as confounders in impact models. Given that the COVID-19 pandemic disrupted baseline data collection, variables were controlled for month of data collection when investigating differences by treatment arm. No formal adjustments were made for multiple comparisons because the two primary outcome variables were treated at $\alpha = 0.05$. All other tests were secondary or exploratory.

The primary outcome variable (grams of fish purchased) was log-transformed due to non-normality. Normality was assessed via histogram inspection and Shapiro-Wilk tests. Effect sizes were estimated using prevalence ratios (for binary outcomes) and log-transformed beta coefficients (for continuous outcomes), with robust standard errors. Interaction effects between FAD and SBC interventions were not formally tested due to insufficient statistical power.

Given that the household survey was conducted over several months, the month of data collection at endline, representing season, was also assessed for confounding. We used mixed-effects linear models for continuous outcomes

**Table 2. Summary of Fish Aggregating Device (FAD) deployment and their impact on catch per unit effort (CPUE) in selected districts of Timor-Leste. The 'CPUE change at EL (kg)' column reflects the change in CPUE at the endline compared to the control site (Leopa), with positive values indicating an increase and negative values a decrease in CPUE following FAD installation.**

| District | Landing site | Randomized to FAD or not | FAD install date | CPUE date range | Number of days district landing was exposed to FAD | CPUE change at EL (kg)[b] |
|---|---|---|---|---|---|---|
| Manufahi[a] | Betano | No FAD | – | – | – | – |
| Liquica | Leopa | No FAD | – | – | – | – |
| Dili | Hera | FAD | 2021/10/01 | 2019/03/04 - 2022/05/03 | 657 | −0.43 |
| Bobonaro | Atabae | FAD | 2021/10/20 | 2019/03/04 - 2022/07/20 | 581 | +0.85 |
| Manatuto[a] | Manatuto | FAD | 2021/10/07 | – | 288 | – |
| Covalima | Suai | FAD | 2021/10/25 | 2020/06/04 - 2021/01/28 | 119 | −0.50 |

[a]Data missing from analysis due to incomplete records. All reported changes in CPUE are statistically significant at the [b]95% confidence level ($p < 0.05$).

and logistic regression for binary outcomes, both with robust standard errors clustered at village level. Confounding was assessed using bivariate models, and confounders were kept in the model using a significance level of $p < 0.10$. Household data were analysed using Stata 14 software (StataCorp). The statistical analysis was done while masked to the treatment arms for as long as it remained feasible (i.e., prior to the presentation of the results).

### 2.8 Spillover analysis

To assess if the effects of the SBC intervention extended beyond the directly targeted households to neighbouring control households, we conducted a spillover analysis using two groups:

(a)*Control with Neighbour SBC*. Households that did not receive the SBC treatment but had neighbours who did.

(b)*Control without Neighbour SBC*. Households that did not receive the SBC treatment and did not have neighbours who received the SBC treatment.

We examined both the difference in days fish was purchased and whether fish was purchased or not for these groups. Additionally, we explored the impact on three key outcome variables; if the household purchased fish (yes/no), the amount of fish purchased, and the amount of fish consumed by women.

Ethics approval for this trial was received from the Timor-Leste *Instituto Nacional de Saúde* (National Health Institute) in December 2020 (1934MS-INS/DE/XII/2020), and the trial was registered with clinical trials.gov (NCT04729829). Statistical analyses were done using R (R Core Team, 2023) and Stata 14 software (StataCorp).

## 3. Results

### 3.1 FAD intervention to enhance fish supply

Our analysis found varying effects of Fish Aggregating Devices (FADs) on CPUE at the study sites (**Table 2**). In the municipalities of Dili and Covalima, a decrease in CPUE was observed compared to the control sites, with median DiD estimates of −0.43 kg and −0.5 kg, respectively. However, in Bobonaro, a significant increase in CPUE of 0.85 kg was found (**Fig 4**).

When FADs became detached (through abrasion or actively sabotaged) within six months of installation, they were replaced. The average duration of the FADs in water was 155.8 (+/-115.1) days (**Table 2**). However, due to a severe weather event followed by poor communication through the COVID pandemic, catch data for the FAD treatment site of Manatuto were lost, and the control site of Manufahi lost all data collected prior to installation, so catch was recorded for 3 treatment sites and 2 control sites, but with complete data available for three treatment sites and one control. The municipality of Manufahi was thus excluded from the analysis.

### 3.2 Household trial profile, recruitment and attrition

This study was implemented by WorldFish in Timor-Leste, from 3 February 2021 to 5 August 2022. In the 24 villages, 755 households participated at baseline (**Table 3**), and 682 participated at endline, representing an attrition rate of 9.7%. Reasons for attrition included respondents not being present in the village during sampling, death, and an unwillingness to respond to an additional survey (**Fig 5**). The trial ended following the completion of the endline survey as intended. However, the total duration of the study was longer than anticipated due to disruptions caused by the COVID19 pandemic.

### 3.3 Impacts of intervention on household fish purchases, fish consumption by women and children

Households in the combined FAD + SBC arm were nearly 2 times as likely as the control arm to report purchasing fish in the previous week (**Table 4**, PR; 1.90, 95% CI; 1.14, 3.20, $p < 0.05$). There were no significant differences in fish purchases in the previous week in the FAD only arm or SBC arm compared to the control (**Table 4**). Among the households

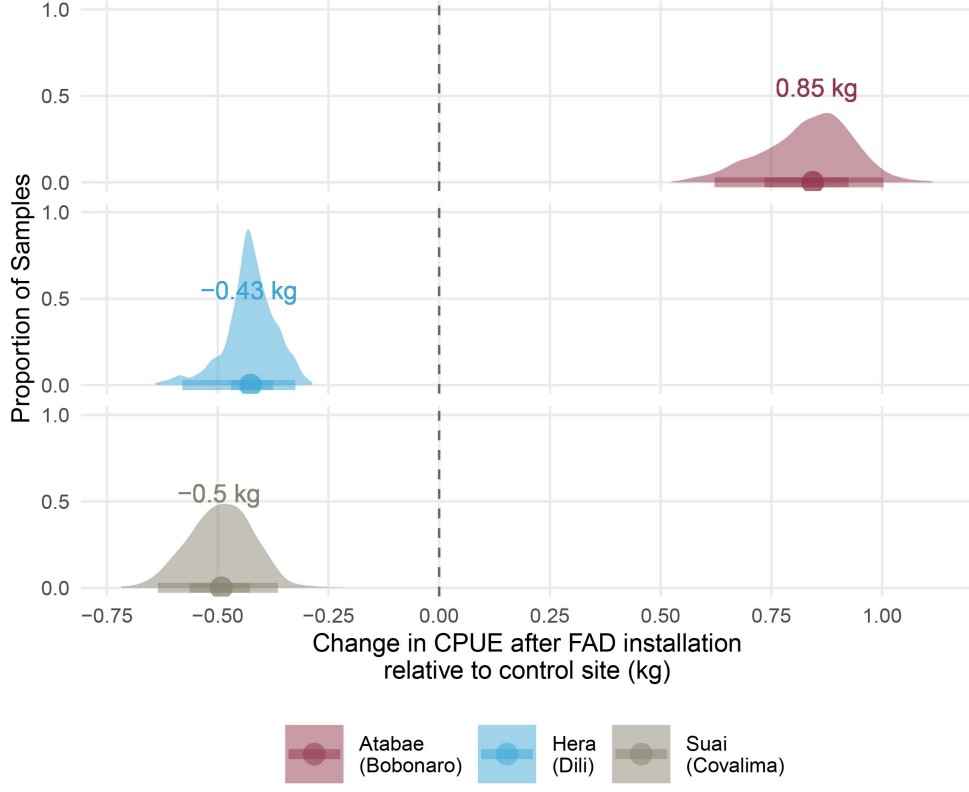

**Fig 4. Density plots of the changes in CPUE following the installation of Fish Aggregating Devices (FADs), compared to the control site, Leopa (Liquica) (represented by the dotted line).** Each plot depicts the distribution of the median CPUE changes derived from 1000 bootstrap resamples at Atabae (Bobonaro), Hera (Dili), and Suai (Covalima) sites, with the median DiD estimate for each site denoted by the number adjacent to the peak of the density. The shaded area beneath each curve represents the 95% confidence interval for the estimates. A vertical dotted line at zero on the x-axis indicates the point of no change in CPUE, which serves as a baseline to evaluate the effect of FAD installation relative to the control site.

that *did* purchase fish in the previous week, the amounts (mean quantity in grams) purchased per household member did not differ significantly by treatment arm (**Table 4**).

Women in the FAD+SBC arms were 4 times as likely to report consuming fish the day prior as compared to women in the control arm (**Table 5**, PR: 4.17, 95% CI 1.88, 9.29; p<0.001). We did not find a significant improvement in child fish consumption by treatment arm. There were no differences in women's dietary diversity by treatment arm, and the percent of women achieving minimum dietary diversity remained low at endline at 8%. It is worth noting that the SBC intervention ended in January 2022, but the household surveys were not conducted until August and September of that same year, indicating that the purchase and consumption habits that resulted as an impact of the intervention were maintained after seven months.

There was no difference by treatment arm of the prevalence of reported knowledge that fish could be fed to children (with proper processing) as early as six months, indicating that knowledge of the benefits of eating fish were already very high at baseline. 33 (6%) respondents reported adding fish powder to the meal of their children between 6–24 months of age. 99% of respondents accurately identified one of the health benefits of feeding fish to children (either improved growth, improved brain power, or better able to fight off sickness) at baseline. We had planned to investigate whether household fish purchases were not mediated by an interaction of the treatment arm and knowledge of the benefits of fish, but preexisting good knowledge of benefits of fish in diets implied there was very little variation in the knowledge of the benefits of fish between baseline and endline.

**Table 3. Unadjusted mean baseline characteristics of households, mothers, and children in the study sample among those living in each of the assigned study arms.**

| Variable | Control (n=65) (no SBC, no FAD) | FAD only (n=319) | SBC only (n=152) | SBC+FAD (n=219) |
|---|---|---|---|---|
| Percentage of households purchasing fish in the previous 7 days[a] | 63% | 38%* | 48% | 49% |
| Average number of days households cooked fish in the previous 7 days (+/-SD) | 1.4 (1.8) | 0.7 (1.1) | 0.9 (1.3) | 1.0 (1.3) |
| Amount of fish purchased by households in previous 7 days (g/household member; geometric mean, 95% CI)[a] | 145.8 (97.7, 217.6) | 64.9 (49.7, 84.9)* | 105.1 (82.5, 133.9) | 69.7 (52.5, 92.6)* |
| Percentage of women who consumed fish in the previous 24 hours | 4% | 7% | 1% | 6% |
| Percentage of children 6–23 months old who consumed fish in the previous 24 hours | 7% | 9% | 4% | 7% |
| Percentage of women who achieved minimum dietary diversity | 8% | 4% | 6% | 3% |
| Percentage of respondents who supported children eating fish from 6 months old | 36% | 39% | 26% | 39% |
| Age of respondent in years (months, SD)[a] | 40.6 (14.9) | 42.5 (16.2) | 40.5 (15.4) | 40.7 (16.0) |
| Percentage of households where HH head has no formal education | 22% | 30% | 30% | 32% |
| Household size (number of occupants, SD) | 4.8 (2.2) | 4.9 (2.3) | 5.0 (2.3) | 4.8 (2.1) |
| Percentage of households with children under five years | 51 | 46 | 46 | 50 |
| Percentage of households with little to no reported hunger[b] | 82 | 77 | 80 | 85 |
| Asset Score (SD) | 1.9 (1.1) | 2.2 (1.4)* | 2.6 (1.4) | 2.9 (1.5)* |
| Percentage of households with at least one member in a VSLA | 26% | 29% | 23% | 37% |
| Percentage of households participating in a nutrition group | 11% | 3% | 3% | 6% |
| Percentage of households participating in a savings group that is not a VSLA | 11% | 15% | 7% | 19% |
| Percentage of households participating in an agricultural cooperative | 8% | 11% | 3% | 10% |
| Percentage of households participating in another community group (support group, NGO-led, etc) | 82% | 76% | 88% | 73% |

[a] Respondent was the person responsible for shopping for the household; [b]Controlling for month of data collection (season), and differences analyzed in log transformed values, back transformed geometric means and 95% CIs presented

*$p < 0.05$ (difference from control arm), based on logistic or linear models with robust standard errors at the village level.

## 3.3 Exploratory analysis

Given the fact that not all the districts randomized to receive a FAD experienced an increase in CPUE as expected, we conducted an exploratory analysis where we recoded the districts that had a FAD but did not experience an increase in catch to the control arm. In this analysis, the 'treatment' arm was the district that experienced a significant increase in catch (Bobonaro). This district had villages in both the FAD only arm and the FAD+SBC only arm; we did not divide them for the exploratory analysis as the reduced sample size would diminish our power too much. The districts for which there was no data on CPUE (Manatuto, Manufahi) were removed from this analysis. The findings for all three primary outcomes, household purchase frequency, amount of fish purchased per household member, and women's fish consumption were assessed using the same methods as described in the primary analysis. In this analysis, the increase in frequency of household fish purchase was not significant, and similar to the primary analysis, there was no difference in the amount of fish purchased per household member (S2 Table). Women in this exploratory analysis were over 8 times as likely to consume fish as women in the control arm (S3 Table, PR: 8.5, 95% CI 1.7, 41.6; $p < 0.05$).

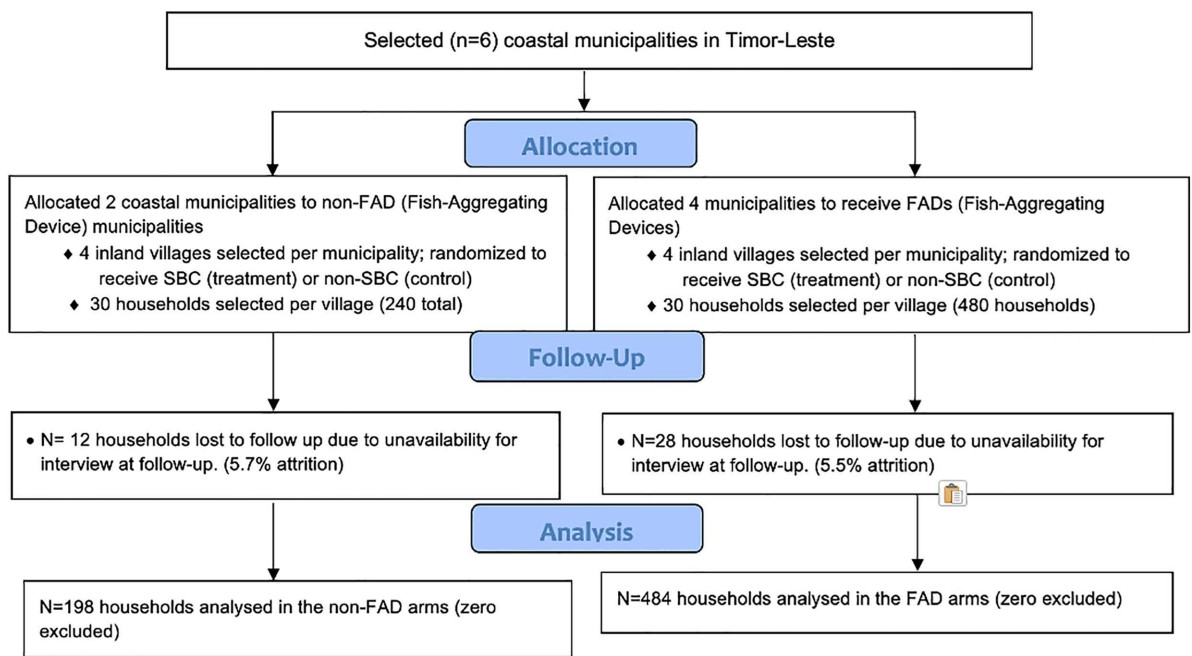

**Fig 5. CONSORT flow diagram detailing the number of households lost to follow up, and the number of households included in final analysis.**

**Table 4. Comparison of differences in household fish purchasing patterns in rural, inland Timor-Leste at endline.**

| Treatment arm | N (households) | Proportion of households purchasing fish in previous 7 days | Prevalence ratio (95% CI)[a] | Mean quantity in grams (CI) of fish purchase/household member/week[b] | Back-transformed β (95% CI)[c] |
|---|---|---|---|---|---|
| Control | 65 | 57% | Ref[d] | 122.20 (74.7, 199.8) | Ref |
| FAD only | 319 | 61% | 1.49 (0.94, 2.37) | 64.5 (50.0, 83.5) | 0.76 (0.39, 1.88) |
| SBC only | 152 | 56% | 1.08 (0.61, 1.91) | 64.4 (40.7, 102.0) | 0.66 (0.25, 1.73) |
| FAD+SBC | 219 | 67% | 1.90 (1.14, 3.20)[e] | 70.2 (51.1, 96.3) | 0.74 (0.29, 1.87) |

[a] Differences in prevalence were assessed by treatment arm using a logistic regression model, controlling for season, household wealth, and fish purchase at baseline (Y/N) with robust standard errors controlling for clustering at the village level.

[b] Geometric means and CI back transformed from logged values presented.

[c] Mean differences analyzed by log-transformed values assessed using a generalized linear model, controlling for household fish purchases at baseline and household wealth with robust standard errors controlling for clustering at the village level.

[d] Ref is short for reference group.

[e] $p < 0.05$.

### 3.4 Spillover analysis

No significant differences in the effect size of the primary outcome variables (household fish purchases and women's fish consumption) were seen between control villages irrespective of their proximity to treatment villages. The mean difference in household fish purchases between control villages and their neighboring villages was minimal and not statistically significant. Similarly, there was no significant difference observed in women's fish consumption between these groups. These findings suggest that the intervention did not produce measurable spillover effects on the primary outcomes in neighboring control villages, indicating that any benefits of the intervention were likely contained within the treated communities.

**Table 5. Fish consumption by women and children in the previous 24 hours at endline.**

| Treatment arm | N (women total) | Women consuming fish in the previous 24 hours | Prevalence ratio (95% CI)[a] | N (children 6–23 months old total) | Children consuming fish in the previous 24 hours[b] |
|---|---|---|---|---|---|
| Control | 46 | 7% | Ref | 11 | 0% |
| FAD only | 209 | 11% | 1.79 (0.53, 5.95) | 60 | 2% |
| SBC only | 115 | 5% | 0.86 (0.37, 1.98) | 37 | 16% |
| FAD + SBC | 155 | 19% | 4.18 (1.88, 9.29)[c] | 39 | 8% |

[a] Differences in prevalence were assessed by treatment arm using a logistic regression model, controlling for household wealth with robust standard errors controlling for clustering at the village level.

[b] We were not able to fit a logistic regression model to look at significant differences in the prevalence of child fish consumption as both the coefficients from the control arm and FAD + SBC arm were dropped because of collinearity.

[c] p < 0.001.

## 3. Discussion

This study addresses a critical gap in food system interventions by testing whether increasing fish supply with nearshore fish aggregating devices (FADs), combined with social and behaviour change (SBC) communication, could improve fish access and dietary intake among nutritionally vulnerable, inland populations in Timor-Leste. We found that the combined FAD + SBC intervention significantly increased both household fish purchase frequency and women's fish consumption. These findings confirm our hypotheses and underscore the importance of integrated, multi-pronged strategies that simultaneously address both supply- and demand-side constraints to dietary improvement. However, the results also reflect the challenges of achieving sustained dietary change in resource-constrained rural populations, highlighting the need for holistic and locally tailored interventions.

The standalone SBC intervention did not result in measurable improvements in fish consumption, indicating that awareness and knowledge alone may be insufficient in contexts where affordability or supply are constrained. Similarly, FADs deployed in isolation led to variable impacts on fish catch per unit effort (CPUE). In particular, two of the four FAD sites experienced declines in CPUE compared to the control, suggesting that FADs are not uniformly effective and should not be assumed to be cost-effective without local feasibility assessments. FADs can make fishing more efficient, by increasing catch while decreasing costs of accessing pelagic fish (in fuel and time spent searching) [43]. In this pilot, installation and maintenance costs were covered by the project, but scaling such interventions would require careful consideration of social, ecological and financial contexts and long-term financing and fisher or government willingness to invest [17]. These supply-side efficiency gains have been shown to be a necessary precursor to realizing dietary benefits even in terrestrial systems, where support of crop production yielded notable improvements in diets and women's and children's health when combined with SBC [44]. Evidence shows that nutrition-sensitive agriculture programs, often focused on increasing production and income of women [45], are more effective at improving child, maternal and household dietary indicators when these integrate nutrition SBC components [45–47]. Our study shows that this might be also the case in nutrition-sensitive fisheries interventions.

### 4.1 Public health implications

Animal-source foods benefit child growth and cognitive development [48–50], and fish are one of the few affordable sources of multiple micronutrients—especially iron, zinc, iodine, and vitamin B12—that are often lacking in low-diversity, staple-based diets [4,6]. In our study, women in FAD + SBC communities were four times more likely to consume fish than those in control areas. Given that maternal diets are a strong predictor of infant nutritional outcomes [51], via both breastfeeding and complementary feeding, women of reproductive age are critical to breaking intergenerational cycles of

malnutrition [52–54]. This has important implications for maternal-child health strategies in small island and coastal developing nations, and broader relevance to global strategies targeting the first 1,000 days of life.

To our knowledge, this is the first randomized controlled trial to assess a nutrition-sensitive fisheries intervention using a factorial design. The significance of this study lies in demonstrating that small-scale fishery technologies can expand the nutritional reach of aquatic foods when paired with behaviour-focused programming. Moreover, anecdotal responses from participants revealed use of preserved forms of fish, such as dried fish and fish powder, especially for infants, indicating that fish preservation and processing may be an important and underexplored pathway for enhancing reach in inland communities [55,56].

## 4.2 Strengths and limitations

The cluster-randomized controlled design of this study allowed for assessment of real-world effectiveness under operational conditions. The use of a validated fisheries monitoring system (Peskas) for CPUE data [57], and a rigorously designed SBC curriculum, enabled a robust linkage of production and consumption outcomes. The adjustment for clustering at the village level further strengthened internal validity. The presence of established village savings and loans associations (VSLAs) as a criterion for village inclusion in the study may have dampened the standalone effect of SBC on fish consumption by only engaging communities that were already exposed to other information related to savings, business, credit etc. However, by asking respondents about other active development projects in their villages during the baseline and endline surveys, we controlled for the influence of other projects on the tested effect. The lack of spillover effects of SBC in control villages underscores the localized impact of the and intervention and therein highlights the challenges of access in Timor-Leste for scaling outcomes through social and commercial networks.

Dietary data were based on self-reporting and subject to recall and social desirability bias, which is a limitation of the study given the lack of external validation. However, if this bias was present, it would have also appeared in the SBC-only treatment, which it did not. Also, while the SBC intervention engaged both women and men, we did not systematically measure changes in intra-household decision-making power, which may have influenced outcomes. While gender and power dynamics are recognised as important influencers of dietary choice and diversity, the intersection of these factors with increased supply and knowledge of nutrition benefits of fish was not explored in this study. However, VSLAs are composed of both men and women members and SBC activities involved all members and encouraged collaborative food purchasing decision-making.

As a highly perishable product, the quality of fish reaching consumers may have differed between upland/inland sites due to factors such as temperature, storage conditions, and travel time (in turn affected by multiple factors). These differences may affect consumer dietary preference. The mention of a mother preparing food with fish powder for her infant highlights a complementary strategy for extending the reach of fish consumption beyond the narrow 30 km range of fresh fish marketing. Furthermore, geographic constraints and small cluster numbers limited statistical power to test interaction effects formally. Lastly, FAD performance varied across sites, supporting evidence of their social and ecological sensitivity and the need for site-specific feasibility assessments [17].

## 4.3 Future research directions

Further interventions should address the low fish intakes among children under two. A comprehensive examination of the pathways linking women's dietary change to household food decision-making and gender roles is warranted. Research is currently underway testing strategies to preserve seasonally abundant fish in different ways, such as fish powder and bottled sardines [56,58], and how these might be integrated into school or maternal health feeding programs [59]. Understanding how local trade, transport, and storage infrastructure influence the durability and equity of impact is also critical for replication and scaling.

## 4.4 Policy and systems implications

The findings of this study have important policy implications both nationally and internationally. On a national level, our results reemphasise the need for policy coherence and support development of policies promoting the access to and integration of fisheries technology with public health and nutrition interventions. Fish is an underused resource with the potential to improve dietary quality in the country [60,61], especially in the rainy season (when crop availability is limited) [62]. Sardines gifted by fishers to children at landing sites, and household gleaning activities conducted predominantly by women and children, already represent a critical source of nutrients for coastal households in Timor-Leste [63]. Long Tom (*Tylosurus* spp.), flying fish (*Cypselurus sp.*), garfish (*Hemiramphus sp.*) and sardines (*Sardinella* spp. and *Amblygaster* spp.) are seasonally abundant marine pelagic fish [64], with peaks in catch occurring mostly during the rainy season, (from November to March), with sardines being in the particularly eco-friendly category of small pelagic fish [6].

Nearshore FADs can act as a cost-effective climate-sensitive solution, making fishing more efficient for food and nutrition security [16] and easing fishing pressure on coral reef populations [65], and bringing economic returns on investment [17]. However, our results suggest that FAD effectiveness is context-specific, and enhancing fish distribution networks is closely coupled with improving roads and cold chains to inland regions. Governments in low- and middle-income countries, like Timor-Leste, must prioritize investments in infrastructure to ensure nutrient flows from fisheries are retained for national consumption where possible [6,66], while encouraging the adoption of nutrition-sensitive interventions to tackle malnutrition in rural areas.

This study contributes to the broader discourse on climate-smart food systems and sustainable development. The findings advocate for the inclusion of small-scale fisheries and innovative fisheries technologies in global discussions on food security, climate change adaptation, and the Sustainable Development Goals (SDGs), particularly SDG 2 (Zero Hunger) and SDG 14 (Life Below Water). Development agencies and international organizations could use these insights to tailor interventions that address both supply and demand-side challenges in food systems across developing regions, specifically in other small island and coastal nations facing similar nutritional challenges.

## Conclusion

This study provides the first randomized controlled trial evidence that combining nearshore fish aggregating devices (FADs) with social and behaviour change (SBC) communication can significantly improve fish consumption among nutritionally vulnerable inland populations. By addressing both supply and demand constraints, the integrated intervention increased household fish purchasing and women's dietary intake—highlighting the potential of small-scale fisheries to contribute to food and nutrition security in small island developing states.

These findings emphasize that supply-side technologies like FADs must be paired with context-specific behavioural strategies to achieve meaningful dietary change. As governments and development partners pursue climate-smart, nutrition-sensitive interventions, the FAD + SBC model offers a scalable approach to enhancing the reach and impact of aquatic foods. These findings offer a potential package of investments for sustainably improving diets, but the problem is complex and additional multi-pronged solutions are needed. Future efforts should focus on sustainability, gender dynamics, and infrastructure investments to extend these benefits across geographies and generations.

## Supporting information

**S1 Table. List of species groupings used for catch monitoring in the Peskas digital monitoring system.**
(DOCX)

**S2 Table. Differences in household fish purchasing patterns in rural, inland Timor-Leste at endline.**
(DOCX)

**S3 Table. Fish consumption by women in the previous 24 hours at endline.**
(DOCX)

**S1 File. Ethics application to Timor-Leste National Institute of Health.**
(PDF)

**S2 File. Registered Report Protocol. Tilley et al (2022) PLoS One.**
(PDF)

**S3 File. CONSORT 2010 Checklist.**
(DOCX)

## Author contributions

**Conceptualization:** Alexander Tilley, Kendra A. Byrd, Katherine Klumpyan, Kelvin Mashisia Shikuku.

**Data curation:** Kendra A. Byrd, Lorenzo Longobardi.

**Formal analysis:** Kendra A. Byrd, Hamza Altarturi, Lorenzo Longobardi.

**Funding acquisition:** Alexander Tilley.

**Investigation:** Alexander Tilley, Joctan Dos Reis Lopes, Mario Gomes, Katherine Klumpyan.

**Methodology:** Alexander Tilley, Katherine Klumpyan, Kelvin Mashisia Shikuku.

**Project administration:** Alexander Tilley, Katherine Klumpyan.

**Software:** Lorenzo Longobardi.

**Supervision:** Alexander Tilley, Katherine Klumpyan.

**Validation:** Kendra A. Byrd, Hamza Altarturi, Joctan Dos Reis Lopes, Mario Gomes.

**Visualization:** Lorenzo Longobardi.

**Writing – original draft:** Alexander Tilley, Kendra A. Byrd, Katherine Klumpyan, Lorenzo Longobardi.

**Writing – review & editing:** Alexander Tilley, Kendra A. Byrd, Gianna Bonis-Profumo, Kelvin Mashisia Shikuku.

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
