## [Decision Letter · Decision Letter 0]

21 Apr 2025

Dear Dr. Tilley,

Thank you for submitting your manuscript to PLOS ONE. After careful consideration, we feel that it has merit but does not fully meet PLOS ONE’s publication criteria as it currently stands. Therefore, we invite you to submit a revised version of the manuscript that addresses the points raised during the review process.

We look forward to receiving your revised manuscript.

Kind regards,

Girma Beressa, MSc, PhD fellow

Academic Editor

PLOS ONE

Journal Requirements:

2. We note that there have been several changes to the author list since the publication of your Registered Report Protocol. Please discuss any changes and ensure that anyone who has contributed to the study but doesn't qualify for authorship has been acknowledged in the Acknowledgements (and their permission to do so obtained). Please confirm in the cover letter that all included authors meet our Authorship criteria. For more details please see https://journals.plos.org/plosone/s/authorship

3. Please expand the acronym “CGIAR” (as indicated in your financial disclosure) so that it states the name of your funders in full.

4. Please note that funding information should not appear in the Acknowledgments section or other areas of your manuscript. We will only publish funding information present in the Funding Statement section of the online submission form. Please remove any funding-related text from the manuscript.

6. We note that Figure 1 in your submission contain map images which may be copyrighted. All PLOS content is published under the Creative Commons Attribution License (CC BY 4.0), which means that the manuscript, images, and Supporting Information files will be freely available online, and any third party is permitted to access, download, copy, distribute, and use these materials in any way, even commercially, with proper attribution. For these reasons, we cannot publish previously copyrighted maps or satellite images created using proprietary data, such as Google software (Google Maps, Street View, and Earth). For more information, see our copyright guidelines: http://journals.plos.org/plosone/s/licenses-and-copyright.

1)  You may seek permission from the original copyright holder of Figure 1 to publish the content specifically under the CC BY 4.0 license.

2) If you are unable to obtain permission from the original copyright holder to publish these figures under the CC BY 4.0 license or if the copyright holder’s requirements are incompatible with the CC BY 4.0 license, please either i) remove the figure or ii) supply a replacement figure that complies with the CC BY 4.0 license. Please check copyright information on all replacement figures and update the figure caption with source information. If applicable, please specify in the figure caption text when a figure is similar but not identical to the original image and is therefore for illustrative purposes only.

7. We notice that your supplementary tables are included in the manuscript file. Please remove them and upload them with the file type 'Supporting Information'. Please ensure that each Supporting Information file has a legend listed in the manuscript after the references list.

**Additional Editor Comments:**

What was the specific type of RCT design?

Authors should recite randomization, allocation, and blinding.

Authors should also specify the type of randomization clearly.

Reviewers' comments:

Reviewer's Responses to Questions

**Comments to the Author**

1. Does the manuscript adhere to the experimental procedures and analyses described in the Registered Report Protocol?

Reviewer #1: Partly

Reviewer #2: Yes

2. If the manuscript reports exploratory analyses or experimental procedures not outlined in the original Registered Report Protocol, are these reasonable, justified and methodologically sound?

Reviewer #1: Yes

Reviewer #2: No

3. Are the conclusions supported by the data and do they address the research question presented in the Registered Report Protocol?

Reviewer #1: Partly

Reviewer #2: Partly

4. Have the authors made all data underlying the findings in their manuscript fully available?

Reviewer #1: Yes

Reviewer #2: Yes

5. Is the manuscript presented in an intelligible fashion and written in standard English?

*PLOS ONE*

Reviewer #1: Yes

Reviewer #2: Yes

Reviewer #1: The report is generally well written and organized. There are several points needing attention or clarification.

The investigators gave much thought to the design of this study as is usually the case when a within cluster randomized approach is attempted. There were 4 interventions including and a control. The effect size was set at 15% with a 0.05 intra cluster correlation. The power was set at 0,80 and presumably a 0.05 alpha level.

1. There should have been some rationale for the intra cluster correlation value.

The analysis approach was quite varied to accommodate endpoint analysis as well as influence of relevant variables on outcomes. The primary outcome variable, the amount of fish purchased per household member over the past week (in grams), was log-transformed to achieve normality.

2. Was this the only non normal concern and how was it tested for normality?

To assess the impact of Fish Aggregating Devices (FADs) on fish catch per unit effort (CPUE), their study implemented a difference-in-difference (DiD) methodology. This method analyses the variation in CPUE before and after FAD installation at the treatment sites and compares these to the changes at a control site where FADs were not deployed. GLM was used for continuous data and the logistic for categorical. Tables 5 and 6 give the primary outcomes for purchase patterns and fish consumption as well as the analysis approach for the specific endpoint. The overall conclusions appear to follow from the analyses performed.

3. However, as per the protocol, any multiple comparisons adjustments were not obvious. This should be discussed in the manuscript text.

There was limited info on confounder influence. Apparently more seasonal or environmental variables were referenced and those variables were included as confounders in the impact evaluation models. If confounders remained significant in the full model (at p<0.05), they were retained. The actual role or significance of expected confounders were not detailed statistically in the logistic or GLM outcomes. The discussion of data management was present, but not detailed in terms of quality control which may not have been the intent in this report. Likewise, the same concern about the detailed discussion for the spillover and mediation analysis which were performed.

4. These concerns should be addressed.

Some important limitations were noted by the investigators such as possible response bias on over consumption and food sharing under reporting.

5. Were any steps taken to validate these responses?

The weather obviously played a role in the loss of information from one region which cannot necessarily be controlled.

For the most part, the statistical design and analysis of this document were well implemented in the report. My comments were minor clarification issues. These concerns noted above about some detail of the analyses lacking such as confounder influence in the models if any, reason for the cluster correlation value, and any multiple comparisons adjustments , if performed, as well as any other issues noted above should be addressed

Reviewer #2: A combined supply and demand intervention increased the frequency of fish consumption by rural women: a randomized, controlled trial

Dear authors

You did a great job and interesting work

The title needs modification

Like this effect of ……….

Greetings

First of all, I appreciate the work you did but I have the following main questions

Why are you targeting women?

What is the novelty of your study since many studies done with similar topics in the country?

In general, the methods part need major revision?

1. Abstract

The background - it would be better if you focused on the severity or magnitudes of the problems or the Gap.

Results: is not written clearly, so would you re-write again

Conclusion: would you write it based on your findings

What is the clinical implication of your findings?

Does the trial registered? where? pls would you include the registration number?

2. Introduction:

Para 1,2 and 3 are redundant as they have no direct relevance to the current objectives of the study, and are not required for the audience of the Journal of PLOS ONE. The introduction section is too short and not clear, and it still is unable to appropriately justify neither the need for this study nor the target age group.

Some of the information provided is excessive and not necessary as the reader quickly becomes inundated with an overabundance of information. Incidentally, the authors should try to provide a more succinct background by truncating/condensing some of the information to provide more specific

In general, the following idea difficult to understand???? need modification, it would be better if you started from definition ---- magnitudes ------ impact ------- etc

What was your hypothesis ? try to include it

3. Methods

Make it clear

What is your sampling frame or unit? How did you use a simple sampling technique to select women?

Do you have references for the operational definition?

How did you control information contamination?

Why didn’t conducted t-test analysis

Have you done difference in difference analysis? If not why?

Did you conduct GEE analysis? If not why

How did you check the normality distribution at baseline and endline ?

Did you calculate the effect size and interaction effect?

4. Results

Check the tables once again

5. Discussion

Need major revision

The discussion needs to be focused on the main objectives. The functional significance of the present results may be elaborated.

Please include the strengths and limitations of the study

What is the clinical implication of your findings? Pls include the clinical implication and future direction of your findings

6. Conclusion

Would you rewrite it based on your findings?

In the ethical consideration part try to include the ethical approval letter number and data.

7. References

Written nicely

**Do you want your identity to be public for this peer review?** For information about this choice, including consent withdrawal, please see our Privacy Policy

Reviewer #1: No

Reviewer #2: No

---

## [Author Response · Author response to Decision Letter 1]

23 Jun 2025

Alll detailed responses are provided in the attached "Response to reviewers" file.

---

## [Editor Report · Decision Letter 1]

14 Jul 2025

Dear Dr. Tilley,

Thank you for submitting your manuscript to PLOS ONE. After careful consideration, we feel that it has merit but does not fully meet PLOS ONE’s publication criteria as it currently stands. Therefore, we invite you to submit a revised version of the manuscript that addresses the points raised during the review process.

**ACADEMIC EDITOR:**

We look forward to receiving your revised manuscript.

Kind regards,

Girma Beressa, MSc, PhD fellow

Academic Editor

PLOS ONE
---

## [Author Response · Author response to Decision Letter 2]

9 Oct 2025

After 4 months I received the following communication from PLOS, so I am resubmitting the ms.

The Detailed responses to reviewer comments are included as a separate file.

Dear Dr. Tilley,

Thank you for your patience as we escalated this case to our Editorial team.

As the original Academic Editor has not responded, our Editorial team is now reassigning them. Please can you resubmit your manuscript as it is, so that the new Editor may issue a decision once they have been assigned.

We apologies for the inconvenience caused by the delay in this process.

Please let us know if you have any questions. We will be happy to help.

Kind regards,

Daniel Davies (he/him)

Peer Review Operations Specialist

---

## [Decision Letter · Decision Letter 2]

28 Nov 2025

Dear Dr. Tilley,

Thank you for submitting your manuscript to PLOS ONE. After careful consideration, we feel that it has merit but does not fully meet PLOS ONE’s publication criteria as it currently stands. Therefore, we invite you to submit a revised version of the manuscript that addresses the minor points raised during the review process, particularly the dew inconsistencies that need clarifications.

We look forward to receiving your revised manuscript.

Kind regards,

Athanassios C. Tsikliras

Academic Editor

PLOS ONE

Journal Requirements:

Reviewers' comments:

Reviewer's Responses to Questions

**Comments to the Author**

1. Does the manuscript adhere to the experimental procedures and analyses described in the Registered Report Protocol?

Reviewer #3: Yes

2. If the manuscript reports exploratory analyses or experimental procedures not outlined in the original Registered Report Protocol, are these reasonable, justified and methodologically sound?

Reviewer #3: Yes

3. Are the conclusions supported by the data and do they address the research question presented in the Registered Report Protocol?

Reviewer #3: Yes

4. Have the authors made all data underlying the findings in their manuscript fully available?

Reviewer #3: Yes

5. Is the manuscript presented in an intelligible fashion and written in standard English?

*PLOS ONE*

Reviewer #3: Yes

Reviewer #3: General comments:

The manuscript explores the important issue of addressing widespread nutritional challenges particularly in rural populations in Timor Leste through a combination of approaches on the supply side and interventions aiming at social behavioural change.

Some of the methodology description is difficult to read and can probably be streamlined. The explanation of the analysis is unnecessarily complicated (e.g. L 355ff).

A few apparent inconsistencies in the text should be clarified:

• it is mentioned repeatedly that FADs are cost-effective and affordable, but given that the FADs in two regions suffered from lower catches than in the reference period without them, this is not quite obvious, especially if fishers would be expected to pay for the installation themselves (presumably, the cost was borne by the project during the testing).

• Only towards the end is there talk about the perishable nature of (fresh) small pelagics marketed inland within 30 km from the landing sites. However, a case of a mother adding fish powder to the food of her infant suggests a dried form of marketing.

• Perhaps more than the great potential to address nutritional deficits the pilot shows the challenges of changing food habits under the constrained conditions of the rural population that requires considerable multi-pronged approaches to have a chance of success at scale. This obviously should not discourage efforts at ensuring nutritious food, particularly for infants and women in child-bearing age, but may also invite a horizon scan for other options to achieve that objective.

Specific comments:

Line 190: correct kg/fisher/hr to kg/fisher x hr

The footnote on page 11 cites publications with authors’ names, not with numbers. The references are missing from the bibliography.

Line 269: SBCC

Line 262: using STATA 13

Line 275: using STATA 14

Line 352: can this be expressed positively? Does this imply SBC made no difference in the women’s knowledge?

Line 358: FAD and SBC only arm.

Line 364f: reference is made to Table S2 and S3: Women are more than 8 times as likely to consume fish? But in Line 395 there is talk of them being 4 times more likely. Evidence needs to be shown and harmonised and the descriptor in Table S2 corrected to: Mean quantity in grams (CI) of fish purchase/household member x weekb.

Line 413: correct grammar

The doc contains 60 references, the layouted version only 47.

Table S1: Decapterus macarellus should be in italics (species level convention)

**Do you want your identity to be public for this peer review?** For information about this choice, including consent withdrawal, please see our Privacy Policy

Reviewer #3: No

---

## [Author Response · Author response to Decision Letter 3]

5 Dec 2025

All reviewer comments are dealt with in the response to reviewers file.

---

## [Editor Report · Decision Letter 3]

30 Dec 2025

A supply and demand intervention increased fish consumption among rural women: a randomized, controlled trial

PONE-D-24-47751R3

Dear Dr. Tilley,

We’re pleased to inform you that your manuscript has been judged scientifically suitable for publication and will be formally accepted for publication once it meets all outstanding technical requirements.

Kind regards,

Athanassios C. Tsikliras

Academic Editor

PLOS One
---

## [Editor Report · Acceptance letter]

PONE-D-24-47751R3

PLOS One

Dear Dr. Tilley,

I'm pleased to inform you that your manuscript has been deemed suitable for publication in PLOS One. Congratulations! Your manuscript is now being handed over to our production team.

Kind regards,

on behalf of

Professor Athanassios C. Tsikliras

Academic Editor

PLOS One